# Exploring Internal Conflicts and Collaboration of a Hospital Home Healthcare Team: A Grounded Theory Approach

**DOI:** 10.3390/healthcare11182478

**Published:** 2023-09-07

**Authors:** Pei-Chun Tai, Shofang Chang

**Affiliations:** 1Department of Quality Center, National Cheng Kung University Hospital, College of Medicine, National Cheng Kung University, Tainan 70403, Taiwan; n147589@mail.hosp.ncku.edu.tw; 2Department of Hospital and Health Care Administration, Chia-Nan University of Pharmacy & Science, Tainan 71710, Taiwan

**Keywords:** home healthcare, hospital home healthcare teams, team collaboration, internal conflict, grounded theory

## Abstract

An aging society is on the rise, leading to a variety of caregiving issues. The Taiwanese government has been implementing a home healthcare integration plan since 2015, aimed at integrating and forming interdisciplinary care teams with medical institutions. This study explores the internal conflict factors among hospital home healthcare team members at a district teaching hospital in Taichung, Taiwan, and it seeks a better collaboration model between them. Semi-structured in-depth interviews were conducted with seven hospital home healthcare team members. Data analysis was based on grounded theory, with research quality relying on the triangulation and consistency analysis methods. The results show that “work overload”, “resource overuse”, “inconsistent assessment”, “limited resources”, “communication cost”, and “lack of incentives” are the major conflicts among the team. This study proposed the following collaboration model, including “identifying the internal stakeholders of a home healthcare team” and “the key stakeholders as referral coordinators”, “patient-centered resource allocation”, and “teamwork orientation”. The study recommends that within a teamwork-oriented home healthcare team, its members should proactively demonstrate their role responsibilities and actively provide support to one another. Only through patient-centered resource allocation and mutual respect can the goal of seamless home healthcare be achieved. The content of the research and samples were approved by the hospital ethics committee (REC108-18).

## 1. Introduction

The world is gradually moving towards an aging society, a transformation accompanied by significant changes in social structure, demographics, and healthcare models. With the increasing elderly population, many incurable chronic illnesses, severe diseases, and disability issues due to health deterioration are emerging gradually. In the past, healthcare services were often fragmented into various domains, with different healthcare professionals operating independently within their respective scopes. This situation led to a blockage of medical information, restricting the transmission and sharing of information between different healthcare specialties and resulting in incomplete treatment plans for patients. However, in order to overcome these issues, an increasing number of healthcare institutions believe that relying solely on single professional services is no longer suitable for the modern healthcare environment. They are shifting towards providing a continuum of care through an “integrated care” approach, and by integrating the knowledge and skill sets of different specialties into healthcare teams, they aim to offer patients more comprehensive treatment plans [1,2,3].

According to information from the World Health Organization (WHO), it is projected that by 2030, globally, one in every six individuals will be aged 60 or above, and the proportion of the population aged 60 and above will increase from 1 billion in 2020 to 1.4 billion. This trend is particularly pronounced in Japan, where it is expected that by 2030, the population aged 60 and above will constitute 30% of the total population [4]. Compared to the US (16.83%) and other OECD countries (17.64%), Japan ranks the highest (28.86%) concerning the percentage of aging population [5]. Such a transformation has made the Japanese government more urgently address various healthcare issues arising from aging. To address these challenges, the Japanese government has integrated hospital medical services and expanded integration into community clinics and social resources. This ensures that patients can receive continuous care from community clinics at home after leaving the hospital [6,7].

Since 1948, the UK has been developing long-term care services, providing patients with a diverse range of medical and social services and accumulating significant experience [8]. In this context, the British Geriatrics Society (BGS) and the Royal College of Physicians (RCP) are part of discussions on home medicine in the Netherlands, which involve a comprehensive interdisciplinary medical team that provides a well-developed integrated care plan, ranging from rehabilitation to palliative care. This medical team includes physicians, physical therapists, speech therapists, occupational therapists, and consultation experts from hospitals. Additionally, a healthcare plan in the UK closely resembles that in the Netherlands [9,10]. Moreover, many patients with chronic diseases are opting for home healthcare plans as an alternative to family medicine [11].

Since 2015, the Taiwanese government has gradually integrated general home care, respiratory home care, and palliative home care into the “Integrated Home Healthcare Plan (iHBMC)”. This program helps chronic patients, disabled people, and individuals walking with difficulty a lot due to its interdisciplinary medical team with different medical specialists [12,13]. Several hospitals in Taiwan have joined iHBMC since patients often require long-term care support upon discharge from the hospital. A discharge planning team is composed of physicians, nurses, dietitians, therapists, therapists, social workers, and others. This team provides integrated care from various medical professions and formulates patient-centered home medical plans based on the needs of both the patient and their family [13].

Home healthcare, as an emerging field, often entails conflicts in the provision of services, and that creates significant challenges during its implementation [7]. Based on past research findings, conflicts can arise during the implementation of home healthcare for various reasons, such as a lack of communication between healthcare professionals and patients, patient characteristics, and insufficient medical equipment, all of which reduce the effectiveness of home healthcare [14,15,16,17]. The aforementioned studies focused mostly on the conflicts of external stakeholders of home healthcare. Fewer of them have investigated conflicts and collaboration within a hospital home healthcare team. According to Freeman, the concept of stakeholders is defined as “any group or individual that can affect or be affected by the achievement of an organization’s objectives, and each of these groups has a stake in the organization” [18]. Internal stakeholders refer to employees working within the organization, including relevant managers, professionals, and non-professionals. When managers identify these key stakeholders, they can ensure that the power and core values of these stakeholders contribute positively to the organization [19]. Evidence has shown that unbalanced power and interests of these stakeholders can promote conflicts [20], and that the conflicts lead to the negative development of a program [21]. Our study explores the conflicts and seek resolutions of collaboration among internal stakeholders within the home healthcare team based on stakeholders’ theory.

Additionally, previous research has typically highlighted the factors contributing to difficulties but has offered few directions for solutions. Consequently, this study aims to fill the gaps and provide resolutions of collaboration, enabling the entire team to execute home medicine more effectively.

## 2. Materials and Methods

### 2.1. Methodology/Study Design

The study was conducted in a semi-structured in-depth interview approach, and it explores the experiences, conflicts, and perceptions of collaboration in a hospital home healthcare team at a regional hospital in Taichung, Taiwan. Purposeful and snowball sampling methods were used to select various roles within the hospital home healthcare team. Prior to the interviews, the research purpose was explained to the participants, and they were required to sign an informed consent form to ensure their understanding of the study’s objectives. After the interviews, the collected data were transcribed verbatim and analyzed using grounded theory. The research findings are then intended to be used as a reference for future practitioners in the field of home healthcare.

### 2.2. Data Collection

This study scheduled interviews in accordance with the working hours of the hospital home healthcare team members. The content of the five interview questions in this study includes: ‘How do members of the hospital home healthcare team interact when performing their duties?’, ‘During this interaction process, what conflicts may arise among them?’, “When conflicts occur, in what ways do team members cooperate to resolve these conflicts?” and “If conflicts cannot be effectively resolved, what do you suggest to improve collaboration?” and “How can the team work to achieve collaboration?” During the interviews, in addition to taking written notes, interviewers were also permitted to use audio recording equipment. Since there are only seven members in the hospital home healthcare team, a total of 7 participants were involved in the interviews for this study (Table 1), encompassing physicians, physical therapists, occupational therapists, home care nurses, regular nurses, and case managers. All interviews took place in the participants’ workplaces during their available time, with durations ranging from approximately 30 to 120 min each. Data analysis was conducted immediately after each interview, and during the analysis, the focus events and key individuals were repeatedly verified to ensure completeness while retaining the most crucial events. When there was repetitive content in the data, this indicated data saturation, and the interviews were discontinued.

### 2.3. Data Analysis Techniques

All the data collected in this study are treated in an anonymous manner. The identity of the participants is represented by abbreviated English letters as participant codes. For example, “Rehabilitation section doctor” was abbreviated as “participant 1,” and this logic was followed for archiving purposes. Based on Strauss’s grounded theory, this study followed a systematic approach. Initially, the verbatim transcripts of the 7 interviews were meticulously reviewed to extract core concepts through the process of conceptualization. Subsequently, these concepts were categorized and linked with participant codes. The categorized concepts were then organized into overarching themes and sub-themes. Throughout this process, the foundational framework of the study was developed by repeatedly confirming the relationships between themes and sub-themes. Lastly, the themes were categorized into the two main subjects of collaboration and conflict. Throughout this compilation process, ongoing comparisons were essential to discern similarities and differences among themes, while extraneous content unrelated to the themes was excluded [22]. Table 2 is an excerpt from the coding process used in our research employing grounded theory.

### 2.4. Quality of the Study

This study employed Guba and Lincoln’s qualitative criteria to assess research rigor [23]. All data were derived from participants’ firsthand real-life experiences, providing reference value and credibility. To enhance the transferability of findings, the interview contents were promptly transcribed into written transcripts, making the research outcomes accessible for use by other home healthcare-related organizations. Additionally, to ensure dependability, the interview outline was designed based on an extensive review of the literature, and similar results can be expected when other researchers explore similar subjects. Moreover, the researchers of this study have undergone training in qualitative research methods, specifically in interview skills and analysis techniques. This training enables them to objectively engage in the research context and focus on neutrality in data and interpretation confirmation. This ensures that the study has evidential confirmability.

In order to enhance the credibility and validity of the research results, this study employed Patton’s triangulation method to verify research rigor. Initially, a comparison was made between the perspectives of healthcare professionals and experts with experience in home healthcare to ascertain whether there were discrepancies in their views regarding collaboration and conflicts within the hospital home healthcare team. Later on, to ensure data completeness, researchers engaged in joint discussions about categories, offering different viewpoints on the research findings. Subsequently, different theories were used to confirm the extent of collaboration and conflicts within the hospital home healthcare team. During the research process, ongoing comparisons were made with the latest information to ensure sufficient data. Additionally, the current work of the team members was verified using relevant existing literature, thorough interviews, and through detailed field notes. These findings will guide future research endeavors [24].

### 2.5. Ethical Consideration

This study has been reviewed and approved by the Research Ethics Committee of Tzu Chi Hospital, a nonprofit organization, with the review number REC108-18.

## 3. Results

### 3.1. The Internal Conflicts of a Home Healthcare Team

The internal conflicts within the home healthcare team are captured in 6 main themes and 14 sub-themes (Table 3).

#### 3.1.1. First Theme: Inconsistent Assessment

The theme points out that among the seven members of the hospital home healthcare team, the most frequent instances of inconsistent assessments are observed between “physicians and therapists” and “physicians and nurses.”

#### 3.1.2. Physicians and Therapists

Therapists mentioned that they follow the home healthcare plans proposed by physicians for patients, but in reality, not every patient actually requires the plan.


*“Of course, if we talk about conflicts, it means that unnecessary treatments are prescribed, and when we request unnecessary home visits, we really don’t know what we can do.”*
(Participant 2)

#### 3.1.3. Physicians and Nurses

Physicians and nurses have differing opinions regarding patients’ home healthcare plans, leading to subsequent resource allocation issues.


*“When we discuss with the Chest Department, they might think that the patient doesn’t have such a need. It’s possible that the doctor subjectively believes that the patient doesn’t have such a need and is unwilling to refer to the Rehabilitation Department. But we have to spend time explaining it to them.”*
(Participant 5)

#### 3.1.4. Second Theme: Overuse of Resources

This theme describes participants’ experiences with resource utilization in home healthcare, and three reasons have led to the “overuse of resources”: “patient families requiring excessive home healthcare”, “careless approval of excess care”, “healthcare professionals inducing care demands”, and “repeated treatments across different care systems (basic home healthcare, advanced care, palliative care)”.

#### 3.1.5. Patient Families Requiring Excessive Home Healthcare

Home healthcare is primarily designed for the convenience of the elderly and patients with mobility challenges. However, participants mentioned that many patients who do not fall into these categories still receive services due to family requests, leading to the overuse of resources.


*“We have encountered situations where the family wants it because they don’t need to take leave and can bring their elderly family members for medical consultations.”*
(Participant 6)

#### 3.1.6. Careless Approval of Excess Care

Participants explained that because home healthcare is a relatively new care model, the development of application criteria is not yet comprehensive. Consequently, when medical professionals carelessly approve excessive services, it leads to the overuse of resources.


*“The criteria for home healthcare are not very strict. If the family raises a certain need, it is considered okay, and the application is approved.”*
(Participant 5)

#### 3.1.7. Healthcare Professionals Inducing Care Demands

Participants indicated that providing home healthcare for patients could generate additional income for themselves. As a result, healthcare professionals might be tempted to induce care demand in order to increase their earnings, leading to resource waste.


*“Some doctors aim to increase their income by taking on a higher number of home healthcare patients. As a result, there are times when we encounter cases that we feel might not be the best fit for home healthcare, yet we still must proceed with the implementation….”*
(Participant 2)

#### 3.1.8. Repeated Treatment among Care Systems

Participants mentioned that the home healthcare plan is divided into three stages based on the patient’s condition: home medical care, home nursing, and palliative care. When a patient seeks treatment across these three stages repeatedly, it becomes challenging for healthcare professionals to manage the patient’s medical records. This situation could potentially lead to resource waste.


*“We try to ensure that the same physician attends to the same patient as much as possible because they would have a better understanding of the patient’s condition. We make an effort to… Also, if a patient recently develops symptoms of dementia, I might request a neurologist to see them because there are still medication issues related to the specialty… But some medications can only be prescribed by specific doctors, so I might need to ask other physicians for assistance…”*
(Participant 4)


*“In home healthcare, there are numerous specific aspects. Some patients require advanced care like tube replacement, but I’m … I’m not responsible for that part. I only know the situation when I’m on site, so there’s nothing I can do…”*
(Participant 5)


*“Actually, the medications we prescribe are limited because you wouldn’t need very potent medications at home. This isn’t our specialty, so… When I can’t control the situation, I directly discuss whether it’s better to transition to palliative home healthcare. We might prescribe morphine or some Fentanyl patches for pain relief, which works better, and then transfer the patient to palliative care. Then… the home care nurse might… also agree with this opinion, and they will refer it to the palliative home healthcare physician. However… palliative care is only provided by general practitioners, but they have restrictions on prescribing medications….”*
(Participant 7)

#### 3.1.9. Third Theme: Limited Resources

This theme, based on participants’ experiences in home healthcare, indicates that limited resources are a result of a decreased budget from the government and insufficient manpower, leading to conflicts within the team.

#### 3.1.10. Decreased Budget from Government

Participants indicated that when the government reduces the budget for home healthcare practitioners, it leads to a decreased willingness within the team to collaborate.


*“In the public sector, they only provide subsidies to physicians and nursing staff. So, if you go out as a team, it incurs quite a lot of costs, including transportation expenses. Consequently, they won’t allocate resources for a pharmacist to join.”*
(Participant 3)

#### 3.1.11. Insufficient Manpower

Participants highlighted that insufficient manpower makes it challenging for them to complete home healthcare tasks within the allotted time. Consequently, team members become less willing to engage in these care responsibilities.


*“Clinical staff handles the clinical aspects, while administrative staff are responsible for tasks like applying for funding and handling subsequent financial matters. If we don’t involve them in the process, we would be overwhelmed with work, wouldn’t we?”*
(Participant 6)

#### 3.1.12. Fourth Theme: Work Overload

The theme emphasizes that work overload is identified as a factor that affects team collaboration. “Inpatients”, “administrative tasks”, and “resources matching” contribute to work overload.

#### 3.1.13. Inpatients

Participants mentioned that apart from home healthcare cases, there are also other clinical patients who require care, such as inpatients.


*“When there are too many cases, we can’t handle them all because there are so many people with needs. But when there are few staff members, we can’t take them all in because we have other clinical patients.”*
(Participant 6)

#### 3.1.14. Administrative Tasks

Participants described that in addition to providing home healthcare for patients, they also have administrative tasks to handle. They believe that this dual responsibility is challenging and should be managed by other specialists.


*“Administrative staff members handle tasks such as applying for funding and subsequent reimbursement. If they don’t involve our team, we will be overwhelmed.”*
(Participant 4)

#### 3.1.15. Resources Matching

Participants explained that an increase in the demand for home healthcare cases results in a higher volume of long-term care resource-matching tasks.


*“Indeed, we have observed that patients from the rehabilitation department require more home healthcare compared to patients from other departments. This could be due to the fact that a significant portion of these patients belong to the neurology department and are generally older in age. They indeed have a higher demand for resources.”*
(Participant 5)

#### 3.1.16. Fifth Theme: Communication Cost

The theme underscores that “communication cost” arises from “unfamiliarity with policies, errors in resource and information sharing”, as well as “inadequate communication between teams and patients.”

#### 3.1.17. Unfamiliarity with Policies, Errors in Resource and Information Sharing

Participants noted that due to home healthcare being a new care model, unfamiliarity with policies among home healthcare professionals leads to resource and communication errors, resulting in conflicts within the team.


*“The government constantly introduces new policies, and sometimes they are experimental. After the trial period, we are still uncertain. For example, we didn’t know that we could apply for assistive devices under home healthcare. In our training, we only knew that assistive devices could be applied for with a disability certificate. But a few years ago, we suddenly learned that assistive devices could be applied for under home healthcare, with certain criteria. The crucial point is that the criteria for applying for assistive devices is different from that of the disability certificate. They are completely separate systems, and no one knows about it.”*
(Participant 1)


*“Are colleagues willing to spend more time understanding this interconnected service? Because many people might only have membership status. But when you play a role similar to a care manager, you need to understand what the policies and regulations are doing, and you need to consider many things for the patients you are caring. It’s not just about being a therapist or a doctor; you also need to be a care manager. Many people are not used to this.”*
(Participant 3)

#### 3.1.18. Inadequate Communication between Teams and Patients

Participants illustrate that when there is a discrepancy in understanding between patients and healthcare professionals, multiple rounds of communication are necessary to reach a consensus. This communication process contributes to an increased workload.


*“I need to pay attention to various matters for the patients, such as what kinds of subsidies they can apply for. When a doctor provides them with a diagnosis, I need to prepare an assessment report for subsidies. When I hand over this assessment report, I have to ensure its accuracy. Sometimes, even if the patient doesn’t qualify for subsidies, they request me to provide the report. Consequently, when they apply themselves, their application might be rejected, and they might think that I didn’t document the report properly.”*
(Participant 3)

#### 3.1.19. Sixth Theme: Lack of Incentives

This theme illustrates that “inconsiderable income” leads to the “lack of incentives” for the hospital home healthcare team.

#### 3.1.20. Inconsiderable Income

Participants indicated that inconsiderable income significantly impacts team collaboration.


*“You’ve been providing these services without compensation for a long time, and it’s difficult to sustain that. When you’re engaged in this type of work, having a funding source is crucial. It helps the staff members achieve a better work-life balance.”*
(Participant 6)

### 3.2. The Internal Collaboration of a Home Healthcare Team

The internal collaboration within the hospital home healthcare team is captured in 4 main themes and 15 sub-themes (Table 4).

#### 3.2.1. First Theme: Key Stakeholders as Referral Coordinators

The theme illustrates that the process of referring cases for home healthcare involves “case managers,” “homecare nurses,” and “physicians from outpatient departments”.

#### 3.2.2. Case Managers

Participants explained that within the hospital home healthcare team, the process of identifying home healthcare cases involves an assessment by the case manager from the discharge planning team to determine whether the patient meets the criteria for home healthcare.


*“We have a discharge preparation service group in our hospital. They have a case manager who receives referrals from all departments, including rehabilitation, respiratory, and neurology.”*
(Participant 3)

#### 3.2.3. Homecare Nurses

Participants mentioned that when cases are referred to home care nurses by case managers, the home care nurses evaluate which home healthcare stage (basic home healthcare, advanced care, palliative care) the patients are suitable for.


*“I would receive information from home care nurses that a new case has been assigned, and the home care nurses would provide me with all the relevant information they have collected.”*
(Participant 1)

#### 3.2.4. Physicians from Outpatient Departments

Participants also expressed that when outpatient physicians from other departments identify patients with a need for home healthcare, they also request assessments for home healthcare services from the discharge planning case managers.


*“The doctor himself noticed during the outpatient visits that every time it was the patient’s son or daughter who came to pick up the prescriptions, and he never saw the patient… It didn’t conform to the normal process of seeing the patient, so he would refer the case to me.”*
(Participant 4)

#### 3.2.5. Second Theme: Internal Stakeholders of a Hospital Home Healthcare Team

This theme underscores the proactive involvement of internal stakeholders, such as “neurology” physicians, “rehabilitation” physicians, and “family medicine” physicians in the activities of the hospital home healthcare team. In terms of patient care, therapists, nurses, and case managers also demonstrate dedicated support for the well-being of the patients.

#### 3.2.6. Neurology, Rehabilitation, and Family Medicine Physicians

Participants mentioned that when there is a shortage of manpower for home healthcare services, physicians are also dedicated and proactive in arranging time to visit patients at their homes to assess their current conditions.


*“Many smaller hospitals lack neurologists who are available for home healthcare visits. However, we are fortunate to have neurologists on our team who are willing to make home visits to patients. As the number of patients has grown, we have been seeking new attending physicians to join us. Eventually, rehabilitation physicians were able to allocate half a day to visit me, which has been very positive and successful.”*
(Participant 7)


*“The cooperation level of physicians in our hospital is quite high. Almost all of the Family Medicine physicians are actively participating.”*
(Participant 4)

#### 3.2.7. Case Managers, OT/PT, and Nurses

Participants mentioned that when case managers from the discharge planning team and the home healthcare nurse have completed the assessment of the patient’s home healthcare needs, including home-based rehabilitation and medical services, the rehabilitation team (OT/PT) proceeds to execute the patient’s rehabilitation plan designed by the physician. Meanwhile, the nurses take care of the nursing aspects.


*“Our rehabilitation team typically consists of physical therapists, occupational therapists, and speech therapists. We begin by understanding the patient’s desired rehabilitation goals and what follow-up examinations they need. Additionally, we determine the prescription medications that need to be prescribed. At this point, we interact with other medical teams. The interaction occurs in two main areas: home-based rehabilitation and home healthcare. For medical care, I discuss with the home healthcare nurses, and for the rehabilitation plan, I communicate with our rehabilitation team.”*
(Participant 1)

#### 3.2.8. Third Theme: Teamwork Orientation

This theme illustrates that when the relevant stakeholders within the team are identified, team members should proactively demonstrate “role positioning” and cultivate a cohesive team spirit to establish a team-oriented approach in home healthcare. Additionally, team members can engage in “post-discharge care meetings” and discussions with patients and their families to determine the direction of care and provide “integrated home healthcare” based on patients’ requirements.

#### 3.2.9. Role Positioning

Participants emphasized that home healthcare professionals should demonstrate role positioning by proactively observing whether patients require home healthcare services and informing them about the services the team can provide.


*“In the realm of home healthcare, there’s a greater need for healthcare professionals to step beyond their usual boundaries. Many times, healthcare professionals, whether therapists, physicians, or nurses, are accustomed to waiting for patients to approach them. However, there’s a need for a change in mindset—to proactively reach out to patients and inform them about how the team can offer assistance.”*
(Participant 3)

#### 3.2.10. Post-Discharge Care Meetings

Participants indicated that through post-discharge care meetings, home healthcare professionals engage in discussions with patients and their families to collectively explore the desired direction for future home healthcare. This process helps facilitate post-discharge care for patients upon returning home.


*“When patients have extended hospital stays and face difficulties with discharge or if their hospitalization period becomes too lengthy, we might assess whether returning home for care poses challenges. In such cases, we communicate with the medical department to determine if involving family members is necessary. We conduct interdisciplinary meetings to discuss this situation, engaging with family members to address concerns they might have when the patient is discharged.”*
(Participant 6)

#### 3.2.11. Mutual Respect

Participants highlighted that when team members have differing opinions, it is important to mutually respect each other’s expertise in order to achieve consensus on treatment goals.


*“We don’t usually have overlaps with physicians from other departments in terms of our clinical duties. However, there are situations where a case that was originally under my care might be temporarily handled by another physician due to medication adjustments or urgent needs. In such cases, if that physician believes it’s better to add or remove a certain medication from the patient’s treatment plan, we would respect their decision. In general, there is mutual respect within the medical profession, and actual conflicts are rare. But if there were, for example, if I strongly believe that a patient should not use a certain type of medication or should be treated differently, we would ask nurses to document it as a note for physicians to be cautious about. This is just a reminder for other physicians and doesn’t involve interfering with their medical practices because physicians are ultimately responsible for their medical decisions.”*
(Participant 1)

#### 3.2.12. Fourth Theme: Patient-Centered Resource Allocation

This theme elucidates that when different professions collaborate to form a hospital home healthcare team, it is crucial to implement a spirit of teamwork and prioritize patient-centered service. Through the “discharge planning team’s assistance” in evaluating patients’ home healthcare needs, resources like home rehabilitation, home therapy, and home healthcare services can be coordinated. Home rehabilitation and therapy are typically managed by “the rehabilitation team”, while home healthcare services are often provided by “integrated outpatient departments” within the hospital, such as family medicine, rehabilitation, and neurology.

#### 3.2.13. Discharge Planning Team

Participants expressed that the discharge planning team assists in evaluating patients’ care needs and linking them with home healthcare resources.


*“The colleagues in the discharge planning service have a nursing background, so they have to assess many things, such as the need for medical equipment and some social service benefits… and also link resources, such as if they need to be transferred to another hospital for treatment.”*
(Participant 3)

#### 3.2.14. Therapist Team

Participants indicated that when patients require home rehabilitation services, the rehabilitation team provides resources based on their needs, such as functional care, assistive device applications, and speech therapy.


*“The collaboration of the rehabilitation team includes physical therapy, occupational therapy, and speech therapy… what goals we hope to achieve and what rehabilitation methods we need to prescribe. It is during this time that we interact with other medical teams.”*
(Participant 1)

#### 3.2.15. Integrated Outpatient Department (OPD)

Participants expressed that when patients with different chronic disease symptoms require professional medical advices, they can seek assistance through the integrated outpatient department (OPD).


*“After being accepted by the post-discharge planning team and applying for long-term care, if the patient is seeing multiple departments, I would help them consolidate, for example, if they are seeing a metabolic department or a cardiology department, I would refer them to the integrated outpatient department.”*
(Participant 6)

## 4. Discussion

Internal conflicts occur where it becomes challenging to maintain cooperation due to differences in opinions and values among internal stakeholders. The findings of this study reveal that in the working environment of home healthcare teams, a shortage of resources such as a lack of economic incentive may have a detrimental impact on team collaboration. Countries with a National Health Service (NHS) system, including Taiwan, the UK, and Japan, can readily implement home healthcare due to robust government regulation [9,10]. However, a single-payer system such as Taiwan may exhibit increased financial inflexibility, making it more susceptible to resource shortages. Conversely, countries with a third-party payer private system like the United States typically adopt a more market-driven approach. In cases of significant market potential, they tend to make substantial investments [25].

When there are communication gaps among team members, it can also easily give rise to conflicts. Moreover, when team members are not familiar with the work purpose and healthcare policies, they may need to repeatedly clarify their job roles, which can decrease work efficiency. These factors, consistent with previous research, can potentially affect the effectiveness of home healthcare services and create an unfavorable atmosphere for teamwork [26,27].

In stakeholders’ theory, Freeman [18] emphasized the “congruent values” between stakeholders. However, our study has found inconsistent assessment as one of the internal conflicts within the hospital home healthcare team. Kadushin and Egan [28] have found ethical conflicts regarding the assessment of mental competence, self-determination, and access to services to be moderately frequent and difficult to resolve. Research has highlighted that inconsistencies among team members in medical assessments can affect patient medication adherence and impede the seamless integration of home and institutional healthcare services [13]. These findings align with the results of our study, indicating that discrepancies in medical assessments can lead to team conflicts.

Through home healthcare integrated care services, healthcare professionals can provide comprehensive assistance to patients after discharge, including medical, psychological, social, and mental support [12,27]. This is especially beneficial for elderly individuals and vulnerable populations with higher daily care needs, as it not only helps improve their health but also delays the decline in physiological function [29]. Our research suggests that, to avoid team conflicts, home healthcare teams should prioritize patient-centered resource allocation in a collaborative manner. Effectively managing and integrating various information enhances team efficiency. Additionally, preventing conflicts and strengthening cooperation among team members can be achieved when individual values align with team goals. The results of this study align with arguments presented in previous literature, emphasizing the positive impact of teamwork-oriented collaboration and shared objectives on work efficiency [1,7,26]. Furthermore, past research has highlighted the necessity of information flow among healthcare professionals [27]. To ensure effective resource and information exchange among home healthcare teams, our study recommends fostering a culture of mutual respect among team members. This culture enables team members to establish mutual trust more easily and collaborate more effectively, actively participating in team discussions to share critical information. Consequently, this approach prevents team conflicts and enhances the overall performance of the home healthcare team.

Based on the research findings on internal conflicts within home healthcare teams presented in this study, we have referred to the suggestions made by Taran. When internal stakeholders hold different role positions, managers can utilize persuasive or educational approaches to coordinate differences with other stakeholders [30]. Freeman [18] also suggested value analysis for stakeholders. These correspond to our study, where we proposed that teams clarify the role definitions and responsibilities of each member through “discharge planning team meetings.” Additionally, we suggest that teams should establish a systematic Standard Operating Procedure (SOP) to ensure that each member can complete tasks in an organized manner according to their job responsibilities. This SOP will serve as a vital reference tool to help maintain consistency and efficiency in work. Furthermore, we encourage team members to actively embrace their roles. This means that each member should wholeheartedly engage and fully utilize their professional knowledge and skills within the team. Such participation helps reduce the occurrence of internal conflicts, as everyone understands their responsibilities and respects the expertise of other members. This proactive involvement also fosters stronger collaborative relationships among team members, leading to more effective joint efforts and enhancing the effectiveness of home healthcare integrated care services.

## 5. Conclusions

Healthcare professionals not only deal with clinical matters but, more importantly, coordinate in assisting patients in obtaining better care in their daily lives. Internal conflicts arise due to occurrences of ‘resource overuse’, ‘inconsistent assessment’, ‘limited resources’, ‘communication costs’ and ‘lack of incentives’.

In order to make home healthcare services more closely aligned with patients’ daily lives, the healthcare team initiates “patient-centered discharge planning meetings” to coordinate home healthcare services. They refer cases to each other and collaborate to reduce workload and balance the work burden. Team members demonstrate their role positioning and work together with mutual respect to help patients find appropriate healthcare resources. Only through patient-centered resource allocation and mutual respect can the goal of seamless home healthcare be achieved.

## 6. Limitations and Recommendations for Future Research

The findings of this study have limited generalizability as the participants are predominantly from the same hospital. Future research could expand the theoretical framework and explore home healthcare teams in other hospitals. Additionally, the interpretation of interview results in qualitative research is not as generalizable as in quantitative research. It is recommended that future studies develop quantitative measurement tools for assessing collaboration and conflicts in home healthcare teams, which would facilitate its application and provide valuable insights for the government and other healthcare agencies in improving home healthcare delivery. Finally, in many countries, religious groups and spiritual life experts play a crucial role in conflict resolution and team integration. Future research could try to explore the feasibility of multidisciplinary professional involvement in conflict resolution.

## Figures and Tables

**Table 1 healthcare-11-02478-t001:** Demographic characteristics of the participants.

Participant	Occupation	Department	Interview Time
Participant 1	Rehabilitation physician	Rehabilitation	48 min
Participant 2	Physical therapist		30 min
Participant 3	Occupational therapist		73 min
Participant 4	Homecare nurse	Discharge planning team	57 min
Participant 5	Nurse		54 min
Participant 6	Case manager		60 min
Participant 7	Neurologist	Neurology	64 min

**Table 2 healthcare-11-02478-t002:** The excerpted coding process of grounded theory in the study.

Categories	Themes	Sub-Themes	Quotes
Conflict	Communication cost	Unfamiliarity with policies, errors in resource and information sharing	*“The government constantly introduces new policies, and sometimes they are experimental. After the trial period, we are still uncertain. For example, we didn’t know that we could apply for assistive devices under home healthcare. In our training, we only knew that assistive devices could be applied for with a disability certificate…”* (Participant 1)
		Inadequate communication between teams and patients	*“I need to pay attention to various matters for the patients, such as what kinds of subsidies they can apply for. When a doctor provides them with a diagnosis, I need to prepare an assessment report for subsidies. When I hand over this assessment report, I have to ensure its accuracy. Sometimes, even if the patient doesn’t qualify for subsidies, they request me to provide the report….”* (Participant 3)
Collaboration	Teamwork orientation	Post-discharge care meeting	*“When patients have extended hospital stays and face difficulties with discharge or if their hospitalization period becomes too lengthy, we might assess whether returning home for care poses challenges. In such cases, we communicate with the medical department to determine if involving family members is necessary….”* (Participant 6)
		Role positioning	*“In the realm of home healthcare, there’s a greater need for healthcare professionals to step beyond their usual boundaries…. However, there’s a need for a change in mindset- to proactively reach out patients and inform them about how the team can offer assistance.”* (Participant 3)
		Mutual respect	*“ …… In general, there is mutual respect within the medical profession, and actual conflicts are rare. But if there were, for example, if I strongly believe that a patient should not use a certain type of medication or should be treated differently, we would ask nurses to document it as a note for physicians to be cautious about. This is just a reminder for other physicians and doesn’t involve interfering with their medical practices because physicians are ultimately responsible for their medical decisions.”* (Participant 1)

**Table 3 healthcare-11-02478-t003:** The internal conflicts of a home healthcare team.

Themes	Sub-Themes
Inconsistent assessment	Physicians and therapists
	Physicians and nurses
Overuse of resources	Patient family requiring excess home care
	Approving excess care carelessly
	Inducing care demands by healthcare professionals
	Repeated treatment among care systems (basic home healthcare, advanced care, palliative care)
Limited resources	Decreased budget from the government
	Insufficient manpower
Work overload	Inpatients
	Administrative work
	Resources matching
Communication cost	Unfamiliarity with policies, errors in resource and information sharing
	Inadequate communication between teams and patients.
Lack of incentives	Inconsiderable income

**Table 4 healthcare-11-02478-t004:** The internal collaboration of a home healthcare team.

Themes	Sub-Themes
Key stakeholders as referral coordinators	Case managers
	Homecare nurses
	Outpatient department (OPD) physicians
Internal stakeholders of a home healthcare team	Neurology
	Family Medicine
	Rehabilitation
	Case manager
	OT/PT
	Nurses
Teamwork orientation	Post-discharge care meeting
	Role positioning
	Mutual respect
Patient-centered resource allocation	Discharge planning team
	Therapist team
	Integrated OPD

## Data Availability

The data is unavailable due to privacy or ethical restrictions.

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
