# Peer review of "Exploring Internal Conflicts and Collaboration of a Hospital Home Healthcare Team: A Grounded Theory Approach"

_healthcare, 2023, doi:10.3390/healthcare11182478_

Round 1

Reviewer 1 Report

Thank you for the opportunity to review this timely and relevant work. While reading the introduction, it was difficult to understand the gap this project filled. There are numerous grammar errors that make reading and understanding the need for this work difficult. My suggestion is to be more specific and direct about what gap in care/gap in the literature this project fills. For example, why does the reader need to know about what Japan does? My suggestion is to start by telling the reader what the problem is....I am guessing there are poor patient outcomes such as increased unplanned hospital visits and/or higher reports of uncontrolled pain in patients who's healthcare team do not communicate effectively? The best advice I have been given to improve my writing is to remember that even academic writing tells a story and the author needs to bring the reader along throughout the story. That doesn't mean you write fiction, but even as someone familiar with the issues you are describing in your manuscript, I am left a little confused. 

The methods section provides too much unnecessary detail and omits details like coding process, member checking, and other qualitative methods that convey rigor. What were the questions that were asked, how did you know you reached saturation, who made those decisions? What member of the team did what and when did you collaborate and work together, what happened if members of the research team didn't agree? I might even omit Tables 1,2, and 3 and instead create a table of exemplar quotes and themes. 

The introduction contained grammar errors and the content was disconnected.

Author Response

Dear Reviewer,

Thank you for your feedback and the opportunity to address your concerns. We've carefully considered your suggestions and have made significant improvements to our paper to enhance clarity and coherence. Kindly allow me to provide detailed responses to the comments and inquiries you have raised. Please see the attachment.

Sincerely,

Shofang Chang

Department of Hospital & Health Care Administration

Chia-Nan University of Pharmacy & Science

Reviewer 2 Report

Dear authors,

Congratulations for researching on a topic of vital importance in our society today. I think this is the line to follow in the coming years.

In the title, at the end you could put the design of the article.

Regarding the bibliography, it would be interesting to leave only the references of the last 5 years, and of the last 10 years when they are methodological or reference documents in the context of the research.

It would also be interesting to present solutions to the results in order to carry out interventions, as proposed in the future lines of research.

Plagiarism is 16.02%, acceptable. However, it would be better if it were below 10%. Revising the introduction and discussion could be achieved.

Author Response

(The authors gave the same response as above.)

Reviewer 3 Report

Generally speaking, the article is well-written. The presented research can complement studies conducted in other centers around the world. However, the paper requires several important corrections.

Firstly, in the Introduction, the authors refer to various healthcare solutions in the UK and Japan but do not mention the approaches adopted in other countries (eg, the USA or some Eastern European countries). This gap needs to be addressed.

Secondly, the term "quality of life" is ambiguous. It would be beneficial for the authors to specify what they specifically mean when using this term (see line 54).

Thirdly, in many countries, chaplains or spiritual life experts play a crucial role in conflict resolution and team integration. The authors do not mention this fact at all (see line 60).

Fourthly, the authors state, "Previous literature has discussed the difficulties in implementing home healthcare practices..." (line 67), but they do not mention which specific literature they are referring to. Thus, the question arises: which authors and literature are being cited in this instance?

Fifthly, the authors overlook important literature on conflicts in the subject matter (eg, Goldie Kadushin, Marcia Egan, Ethical Dilemmas in Home Health Care: A Social Work Perspective, Health & Social Work, Volume 26, Issue 3, August 2001, Pages 136-149, https://doi.org/10.1093/hsw/26.3.136). It is essential to supplement the existing literature.

Finally, the authors fail to explain why only 7 participants were included in the study (See: Data Collection). It is doubtful that such a small number of participants could be representative of the problem presented in the article. The authors should, at the very least, provide an explanation for limiting the study to only 7 individuals.

Author Response

Dear Reviewer,

Thank you for your feedback and the opportunity to address your concerns. We've carefully considered your suggestions and have made significant improvements to our paper to enhance clarity and coherence. Kindly allow me to provide detailed responses to the comments and inquiries you have raised. Please see attachment.

Sincerely,

Shofang Chang

Department of Hospital & Health Care Administration

Chia-Nan University of Pharmacy & Science

Round 2

Reviewer 1 Report

The revisions add rigor to the project, especially a qualitative work which we need more of. Well done

Author Response

Dear reviewer, 

Thank you very much for your for your kind words regarding my submitted manuscript. Your insightful comments and positive feedback have inspired and motivated me to continue my research efforts in this field.

Best regards,
Shofang Chang
Associate Professor
Department of Hospital and Health Care Administration,
Chia-Nan University of Pharmacy & Science

Reviewer 3 Report

I have no more comments. I accept the article in its current form.

Author Response

Dear reviewer, 

Thank you for the kind encouragement. Your constructive feedback has undoubtedly played a crucial role in refining the quality of the manuscript, and I am truly grateful for your valuable input.

Best regards,
Shofang Chang
Associate Professor
Department of Hospital and Health Care Administration,
Chia-Nan University of Pharmacy & Science